# Projection-Free Methods for Solving Nonconvex-Concave Saddle Point Problems

**Morteza Boroun**[*]**, Erfan Yazdandoost Hamedani**[*]**, Afrooz Jalilzadeh**[*]
[*] Department of Systems & Industrial Engineering
The University of Arizona
Tucson, AZ 85721
morteza@arizona.edu, erfany@arizona.edu, afrooz@arizona.edu

## Abstract

In this paper, we investigate a class of constrained saddle point (SP) problems where the objective function is nonconvex-concave and smooth. This class of problems has wide applicability in machine learning, including robust multi-class classification and dictionary learning. Several projection-based primal-dual methods have been developed to tackle this problem; however, the availability of methods with projection-free oracles remains limited. To address this gap, we propose efficient single-loop projection-free methods reliant on first-order information. In particular, using regularization and nested approximation techniques, we propose a primal-dual conditional gradient method that solely employs linear minimization oracles to handle constraints. Assuming that the constraint set in the maximization is strongly convex, our method achieves an $\epsilon$-stationary solution within $\mathcal{O}(\epsilon^{-6})$ iterations. When the projection onto the constraint set of maximization is easy to compute, we propose a one-sided projection-free method that achieves an $\epsilon$-stationary solution within $\mathcal{O}(\epsilon^{-4})$ iterations. Moreover, we present improved iteration complexities of our methods under a strong concavity assumption. To the best of our knowledge, our proposed algorithms are among the first projection-free methods with convergence guarantees for solving nonconvex-concave SP problems.

## 1 Introduction

Let $(\mathcal{X}, \|\cdot\|_{\mathcal{X}})$ and $(\mathcal{Y}, \|\cdot\|_{\mathcal{Y}})$ be finite-dimensional, real normed spaces with the corresponding dual spaces denoted by $(\mathcal{X}^*, \|\cdot\|_{\mathcal{X}^*})$ and $(\mathcal{Y}^*, \|\cdot\|_{\mathcal{Y}^*})$, respectively. In this paper, we study a saddle point (SP) problem of the following form:

$$\min_{x \in X} \max_{y \in Y} \ \mathcal{L}(x, y), \tag{1}$$

where $\mathcal{L} : \mathcal{X} \times \mathcal{Y} \to \mathbb{R}$ is possibly nonconvex in $x$ for any $y \in Y$ and concave in $y$ for any $x \in X$ with certain differentiability properties (see Assumption 2.6); moreover, $X \subseteq \mathcal{X}$ and $Y \subseteq \mathcal{Y}$ are convex and compact sets. Such a problem has a broad range of applications including robust optimization [4], reinforcement learning [11], and adversarial learning [17], to name just a few.

There has been vast research conducted on developing algorithms for solving SP problems [37, 41, 8, 42, 18]. Despite the extensive studies conducted on the development of algorithms to solve problem (1) a common drawback of these methods is their reliance on costly operations involving the projection onto constraint sets. In the optimization literature, the shortcomings associated with projection-based techniques prompted the emergence and advancement of projection-free approaches. Namely, Frank Wolfe-based algorithms [14] are first-order methods designed for minimizing smooth constrained optimization problems where instead of projection onto a constraint set only a linear minimization

over the set needs to be solved. Such an operation can significantly reduce the computational cost of the algorithm as in the case of nuclear-norm ball constraints.

Nonconvex SP problems have already been extensively investigated in the literature and various methods have been proposed for different settings. Namely, the vanilla projected gradient descent ascent has been shown to achieve iteration complexity $\mathcal{O}(\epsilon^{-2})$ for finding an $\epsilon$-stationary of nonconvex-strongly concave problem [27] matching the lower bound in this setting. However, this method fails to address nonconvex-concave problems, therefore, many researchers studied various smoothing [51] and acceleration [40] techniques to ensure a convergence rate guarantee. Most of these methods are built upon the idea of solving regularized subproblems inexactly resulting in a multi-loop method that can pose implementation challenges. To avoid such a complication, single-loop primal-dual methods have gained popularity. Recently, a unified projected primal-dual gradient using the regularization technique has been studied [49]. The proposed method can achieve the complexities of $\mathcal{O}(\epsilon^{-2})$ and $\mathcal{O}(\epsilon^{-4})$ for nonconvex-strongly concave and nonconvex-concave settings. While the studies on primal-dual projected gradient-based methods are rich and fruitful, less is known about projected-free methods for SP problems. With only a handful of studies focusing on (strongly) convex-(strongly) concave problems [16, 46, 9], no projection-free method for solving nonconvex SP problems is currently available, to the best of our knowledge. Moreover, recently there has been a growing interest in developing algorithms for solving bilevel optimization problems [43, 20, 13] which subsumes SP problems as a special case. However, it is important to note that most of the existing methods for solving bilevel optimization problems consider an additional assumption on the lower-level objective function satisfying strong convexity or Polyak Lojasiewicz (PL) condition [20, 2, 31]. These assumptions in the context of SP problems translate into strong concavity or PL condition for $\mathcal{L}(x, \cdot)$ which cannot handle merely concave setting considered in this paper. Therefore, in this paper, we focus on developing projection-free methods for solving (1). In particular, our main research question is

*Can we develop a one-sided or fully projection-free primal-dual method with a convergence rate guarantee for solving nonconvex-(strongly) concave SP problem in* (1)*?*

Our response to this question is affirmative, and next we outline the key contributions of our work.

## 1.1 Contribution

In this paper, we consider a broad class of constrained SP problems where the objective function is nonconvex-concave and smooth. Motivated by the pressing need for developing projection-free algorithms from an application perspective as well as the lack of efficient methods with theoretical guarantees, we propose primal-dual projection-free methods for solving nonconvex-concave SP problems. Using regularization and nested approximation techniques we first develop a single-loop primal-dual method that solely uses linear minimization oracle (LMO) to find a direction for both primal and dual steps. Next, assuming that the projection oracle (PO) is available for the maximization component of the objective function, we develop a one-sided projection-free method with regularized projection gradient ascent. Our main theoretical contributions are summarized below.

**1)** Assuming the constraint set in the maximization component is strongly convex, we show that our proposed primal-dual conditional gradient method that employs LMOs for both variables achieves an $\epsilon$-stationary solution within $\mathcal{O}(\epsilon^{-6})$ iterations. To the best of our knowledge, this is the first complexity result for primal-dual methods with only LMOs for nonconvex-concave SP problems. Moreover, when the objective function is nonconvex-strongly concave, our proposed method achieves $\epsilon$-primal and $\epsilon$-dual gaps within $\mathcal{O}(\epsilon^{-4})$ and $\mathcal{O}(\epsilon^{-2})$ iterations, respectively.

**2)** Considering the case where projection onto the constraint set of the maximization component is easy to compute, we demonstrate that our one-sided projection-free method achieves an $\epsilon$-stationary solution within $\mathcal{O}(\epsilon^{-4})$ iterations matching the best-known results for single-loop projection-based primal-dual methods. For nonconvex-strongly concave setting, the iteration complexity improves to $\mathcal{O}(\epsilon^{-2})$.

## 1.2 Related work

SP problems have been subject to extensive studies, primarily in the context of the convex-concave setting [37, 41, 8, 18]. However, the practical applications of SP problems in nonconvex settings,

particularly in the context of robust optimization and adversarial learning have sparked a demand for a thorough investigation and further exploration. Next, we briefly discuss the state-of-the-art methods in the literature for nonconvex-(strongly) concave problems. We divide our review into two sections of projected gradient-based and projection-free-based algorithms.

**Projected gradient based algorithms:** Various algorithms have been proposed for nonconvex-concave setting under different assumptions [30, 7, 40, 52, 6, 3]. Authors in [44] proposed a double loop algorithm to solve nonsmooth weakly convex-concave with a complexity of $\tilde{\mathcal{O}}(\epsilon^{-6})$. In the pursuit of more efficient solutions, Triple-loop inexact proximal point methods with the faster convergence rate of $\tilde{\mathcal{O}}(\epsilon^{-3})$ are developed in [48, 23]. Additionally, single-loop algorithms for addressing nonconvex-concave problems were introduced by [29] and [32] where they achieved the iteration complexities of $\tilde{\mathcal{O}}(\epsilon^{-6})$ and $\tilde{\mathcal{O}}(\epsilon^{-4})$, respectively. More recently, authors in [33] proposed optimistic gradient descent ascent and extra-gradient methods and obtained a convergence rate of $\mathcal{O}(\epsilon^{-6})$. Employing Nesterov's smoothing technique [38], the iteration complexity of $\mathcal{O}(\epsilon^{-4})$ was obtained by [51] for general nonconvex-concave problem. In addition, [50] achieved the same convergence rate with linear coupled equality or inequality constraints. Recently, [49] proposed an alternating gradient projection (AGP) algorithm where it utilizes gradient projection for updating $x$ and $y$ and it achieves the suboptimal complexity of $\mathcal{O}(\epsilon^{-4})$. All the aforementioned methods rely on projection onto the constraint sets.

**Projection-free based algorithms:** There has been a surge of interest in the application of projection-free methods within the machine learning and optimization problems [24, 15, 25, 26]. Despite the widespread application of projection-free methods such as the Frank Wolfe (FW) algorithm, there have been limited studies that examine their use in the context of SP problems [1, 19, 47, 16, 46, 9, 22]. In [19] a projection-free algorithm for convex-concave setting is proposed and achieved the complexity of $\mathcal{O}(\epsilon^{-2})$ LMO calls. This algorithm was subsequently refined by [47] with the development of a parallelizable algorithm. Gidel et al. [16] presented the first convergence result for an FW-type algorithm for nonbilinear SP problems. Assuming that the SP solution lies within the interior of constraint sets, the proposed method achieves a linear rate for strongly convex-strongly concave SP problems. Moreover, the method achieves the same complexity for convex-concave setting, assuming that the constraint sets are strongly convex and gradients of the objective function are uniformly lower-bounded. Roy et al. [46] advanced a projection-free algorithm for the same setting and introduced a variant of the FW algorithm that is particularly suited for the nonstationary stochastic SP problems. Additionally, the authors in [9] proposed a four-loop projection-free algorithm which can be viewed as a variant of Mirror-Prox in which proximal subproblems are solved inexactly using conditional gradient sliding [25] for convex-strongly concave problems. Considering a nonconvex-concave setting, Nouiehed et al. [39] proposed a multi-loop one-sided projection-free primal-dual method that achieves the complexity of $\tilde{\mathcal{O}}(\epsilon^{-3.5})$ to find an $\epsilon$-game stationary solution. Notably, in their method, at each iteration $k$ the minimization subproblem needs to be computed over the set $\{x \in \mathcal{X} \mid x + x_k \in X, \|x\| \leq 1\}$. Kolmogorov et al. [22] developed a two-loop one-sided projection-free primal-dual method for solving bilinear convex-concave SP problems where LMO is solely used for computing the minimization variable. The proposed method achieves an iteration complexity of $\mathcal{O}(\epsilon^{-1})$. The summary of related work and their comparison are presented in Table 1.

## 1.3 Motivating Examples

**Example 1. Robust Multiclass Classification:** Consider a multiclass classification task for a given training dataset $\mathcal{D}^{tr} = \{(\mathbf{a}_i, b_i)\}_{i=1}^{n}$ where $\mathbf{a}_i \in \mathbb{R}^d$ denotes the feature vector of the $i$-th sample and $b_i \in \{1, \ldots, k\}$ is the corresponding label. The goal is to find a linear predictor with parameter $\boldsymbol{\theta} = [\theta_1^\top, \theta_2^\top, \ldots \theta_k^\top] \in \mathbb{R}^{k \times d}$ that is able to predict the labels for an unlabeled dataset $\mathcal{D}^{test} = \{\mathbf{a}_j\}_{j=1}^{n'}$. In particular, given a new data point $\hat{\mathbf{a}} \in \mathcal{D}^{test}$, the corresponding label can be predicted via $\hat{b} = \arg\max_{j=1}^{k} \theta_j^\top \hat{\mathbf{a}}$. It has been shown that [12] the linear predictor $\boldsymbol{\theta}$ can be found by minimizing a multinomial logistic loss function, i.e., $\ell_i(\boldsymbol{\theta}) = \log\left(1 + \exp(\sum_{j \neq b_i}(\theta_j^\top \mathbf{a}_i - \theta_{b_i}^\top \mathbf{a}_i))\right)$, over a nuclear norm constraint, i.e., $X = \{\boldsymbol{\theta} \mid \|\boldsymbol{\theta}\|_* \leq r\}$ for some $r > 0$. On the other hand, in many applications where safety and reliability are crucial distributionally robust optimization offers a promising approach for a more robust and reliable prediction that can be used in classification tasks to capture uncertainty in data distribution [35, 34]. This can be achieved by considering a worst-case performance of non-parametric uncertainty set on the underlying data distribution. This leads to a min-max formulation where the maximization is taking over an uncertainty set $Y$. For

Table 1: Comparative analysis of optimization schemes for convex-strongly concave (C-SC), convex-concave (C-C), nonconvex-strongly concave (NC-SC) and nonconvex-concave (NC-C) problems, featuring different oracles: Linear Minimization Oracle (LMO) and Projection/Proximal Oracle (PO). $\tilde{O}(\cdot)$ denotes $\mathcal{O}(\cdot)$ up to a logarithmic factor. In the column of "additional assumption", the abbreviation SC stands for strongly convex. $^*$ The complexities are presented for $\epsilon$-primal and $\epsilon$-dual, respectively.

| Setting | Ref | Non-bilinear | Oracle | # of loops | Addt'l Assump. | Rate |
|---|---|---|---|---|---|---|
| C-SC | Chen et al. [9] | ✓ | LMO-LMO | 4 | None | $\tilde{\mathcal{O}}(\epsilon^{-2})$ |
| C-C | Gidel et al. [16] | ✓ | LMO-LMO | 1 | $X$ and $Y$ are SC sets | $\mathcal{O}(\log \epsilon^{-1})$ |
| | Kolmogorov et al. [22] | ✗ | LMO-PO | 2 | $X$ is SC set | $\mathcal{O}(\epsilon^{-1})$ |
| NC-SC | Xu et al. [49] | ✓ | PO-PO | 1 | None | $\mathcal{O}(\epsilon^{-2})$ |
| | **CG-RPGA** (Theorem 4.2) | ✓ | LMO-PO | 1 | None | $\mathcal{O}(\epsilon^{-2})$ |
| | **R-PDCG** (Theorem 4.4) | ✓ | LMO-LMO | 1 | $Y$ is SC set | $\mathcal{O}(\epsilon^{-4}), \mathcal{O}(\epsilon^{-2})^*$ |
| NC-C | Xu et al. [49] | ✓ | PO-PO | 1 | None | $\mathcal{O}(\epsilon^{-4})$ |
| | **CG-RPGA** (Theorem 5.1) | ✓ | LMO-PO | 1 | None | $\mathcal{O}(\epsilon^{-4})$ |
| | **R-PDCG** (Theorem 5.3) | ✓ | LMO-LMO | 1 | $Y$ is SC set | $\mathcal{O}(\epsilon^{-6})$ |

instance, $Y = \{y \in \Delta_n : V(y, \frac{1}{n}\mathbf{1}_n) \leq \rho\}$ is considered in different papers such as [34], where $\Delta_n$ is an $n$-dimensional simplex set, and $V(Q, P)$ denotes the divergence measure between two sets of probability measures $Q$ and $P$. Therefore, distributionally robust multiclass classification can be formulated as the following SP problem [9]:

$$\min_{\boldsymbol{\theta} \in X} \max_{y=[y_i]_{i=1}^n \in Y} \sum_{i=1}^{n} y_i \ell_i(\boldsymbol{\theta}). \tag{2}$$

This problem can be readily cast as (1) by defining $\mathcal{L}(\boldsymbol{\theta}, y) = \sum_{i=1}^n y_i \ell_i(\boldsymbol{\theta})$. The constraint of the maximization is the intersection of simplex set and divergence measure constraints. Indeed, one can relax the simplex constraint using the splitting technique and Fenchel duality. The resulting equivalent saddle point problem has a maximization constraint of $Y = \{y : V(y, \frac{1}{n}\mathbf{1}_n) \leq \rho\}$. which is only described by the divergence measure constraint. In some popular examples such as the Pearson Chi-square divergence, i.e., $V(y, \mathbf{1}_n/n) = \|ny - \mathbf{1}_n\|^2$, $Y$ satisfies the assumption of strongly convex constraint set.

**Example 2. Dictionary Learning:** Dictionary learning aims to acquire a succinct representation of the input data extracted from a large dataset. Given an input dataset $\mathbf{A} = [a_1, \dots, a_n] \in \mathbb{R}^{m \times n}$, our primary goal is to find a dictionary $\mathbf{D} = [d_1, \dots, d_p] \in \mathbb{R}^{m \times p}$ that can accurately approximate the data through linear combinations. Dictionary learning problem can be formulated in nonconvex optimization form [45]:

$$\min_{\mathbf{D} \in \mathbb{R}^{m \times p}} \min_{\mathbf{C} \in \mathbb{R}^{p \times n}} \|\mathbf{A} - \mathbf{DC}\|_F^2, \quad \text{s.t.} \quad \|\mathbf{C}\|_* \leq r; \ \|d_j\|_2 \leq 1, \forall j \in \{1, \dots, p\}, \tag{3}$$

where $\mathbf{C} \in \mathbb{R}^{p \times n}$ denotes the coefficient matrix. In various learning scenarios, such as lifelong learning, the learner engages in a sequential series of tasks, aiming to accumulate knowledge from previous tasks in order to enhance performance in subsequent tasks. Suppose that we have learned a dictionary $\mathbf{D} \in \mathbb{R}^{m \times p}$ and its corresponding coefficient matrix $\mathbf{C} \in \mathbb{R}^{p \times n}$ for the dataset $\mathbf{A}$. Given a new dataset $\mathbf{A}' \in \mathbb{R}^{m \times n'}$, we aim to refine a new dictionary $\mathbf{D}' \in \mathbb{R}^{m \times q}$ such that it still provides an accurate representation of old dataset $\mathbf{A}$ with the learned coefficient matrix $\mathbf{C}$. Consequently, this problem 3 can be written as the following non-convex problem with a convex nonlinear constraint:

$$\min_{\mathbf{D}'} \min_{\mathbf{C}'} \|\mathbf{A}' - \mathbf{D}'\mathbf{C}'\|_F^2, \ \text{s.t.} \ \|\mathbf{A} - \mathbf{D}'\tilde{\mathbf{C}}\|_F^2 \leq \delta; \ \|\mathbf{C}'\|_* \leq r; \ \|d_j'\|_2 \leq 1, \forall j \in \{1, \dots, q\}, \tag{4}$$

where $\delta > 0$ denotes the user-predefined accuracy for the representation of the old dataset and $\tilde{\mathbf{C}} \in \mathbb{R}^{q \times n}$ is the extension of $\mathbf{C}$ by adding $q - p$ columns of zeros. This problem can be formulated equivalently as an SP problem (1) using a Lagrangian duality. In particular, it can be cast as (1) by setting $\mathcal{L}((\mathbf{D}', \mathbf{C}'), y) = \|\mathbf{A}' - \mathbf{D}'\mathbf{C}'\|_F^2 + y(\|\mathbf{A} - \mathbf{D}'\tilde{\mathbf{C}}\|_F^2 - \delta)$, $X = \{(\mathbf{D}', \mathbf{C}') \mid \|\mathbf{C}'\|_* \leq r, \ \|d_j'\|_2 \leq 1, \forall j \in \{1, \dots, q\}\}$, and $Y = \mathbb{R}_+$. Note that $\mathcal{L}$ is nonconvex-concave and smooth. Moreover, due to the existence of a Slater point for problem (4) a dual bound, i.e., $\|y\| \leq B$ for some $B > 0$, can be constructed efficiently as suggested in [36, 18] to ensure boundedness of set $Y$.

Other than min-max formulation, the nuclear-norm constraint in the above examples makes the problem computationally demanding for projection-based algorithms. Therefore, it is imperative to address this challenge by developing and utilizing projection-free methods, which can effectively overcome the computational limitations associated with these problems.

## 2 Preliminaries

In this section, we outline the notations and required assumptions that we need for the analysis of our proposed methods as well as some important definitions.

**Notations.** Given two sets $X$ and $Y$, $X \times Y$ denotes the Cartesian product of $X$ and $Y$. The lower-case $x$ and $y$, as well as their accented variants, are reserved for vectors in the space of $\mathcal{X}$ and $\mathcal{Y}$, respectively; moreover, the letter $z$ will be used to indicate the concatenation of those vectors, i.e., $z \triangleq [x^\top \ y^\top]^\top$. The upper-case $X$ and $Y$ are reserved for constraint sets; moreover, $Z$ will be used to indicate their Cartesian product. Given a differentiable function $\mathcal{L}(x, y)$, $\nabla_x \mathcal{L}(x, y)$ and $\nabla_y \mathcal{L}(x, y)$ denote the partial derivatives of $\mathcal{L}$ with respect to $x$ and $y$, respectively. Given a set $X$, $\mathbb{1}_X(\cdot)$ denotes the indicator function. Given a finite-dimensional normed vector space, $\|\cdot\|_p$ for some $p \geq 1$ denotes the $\ell_p$-norm.

### 2.1 Oracle Description

In this paper, we will utilize the following oracles to acquire first-order information and solution of structured subproblems for the proposed algorithms in different settings.

- Given $(x, y)$, first-order oracle (FO) returns $\nabla_x \mathcal{L}(x, y)$ and $\nabla_y \mathcal{L}(x, y)$.
- Given a vector $\bar{x} \in \mathcal{U}$ and a convex and compact set $U \subseteq \mathcal{U}$, linear minimization oracle (LMO) returns a solution of $\min_{x \in U} \langle \bar{x}, x \rangle$.
- Given a vector $\bar{x} \in \mathcal{U}$ and a convex and compact set $U \subseteq \mathcal{U}$, projection oracle (PO) returns the solution of $\min_{x \in U} \|x - \bar{x}\|_{\mathcal{U}}$.

Based on the above oracles, we define some gap functions to measure the quality of the solution. More specifically, when the proposed algorithm uses LMO we use the following definition.

**Definition 2.1** (Gap function for LMO)**.** The stationary gap function $\mathcal{G}_X : Z \to \mathbb{R}$ for the minimization part of problem (1) is defined as $\mathcal{G}_X(\bar{x}, \bar{y}) \triangleq \sup_{x \in X} \langle \nabla_x \mathcal{L}(\bar{x}, \bar{y}), \bar{x} - x \rangle$. Similarly, for the maximization part of problem (1) the stationary gap function $\mathcal{G}_Y : Z \to \mathbb{R}$ is defined as $\mathcal{G}_Y(\bar{x}, \bar{y}) \triangleq \sup_{y \in Y} \langle \nabla_y \mathcal{L}(\bar{x}, \bar{y}), y - \bar{y} \rangle$. Moreover, we define $\mathcal{G}_Z(\bar{x}, \bar{y}) \triangleq \mathcal{G}_X(\bar{x}, \bar{y}) + \mathcal{G}_Y(\bar{x}, \bar{y})$.

In the case where the proposed method uses PO for the maximization part of the objective function of (1), we use the same gap function $\mathcal{G}_X$ for the minimization component while employing the following gap function for the maximization component.

**Definition 2.2** (Gap function for PO)**.** The stationary dual gap function $\mathcal{G}_Y : Z \to \mathbb{R}$ corresponding to the maximization part of problem (1) is defined as $\mathcal{G}_Y(\bar{x}, \bar{y}) \triangleq \frac{1}{\sigma} \|y - \mathcal{P}_Y(y + \sigma \nabla_y \mathcal{L}(\bar{x}, \bar{y}))\|_{\mathcal{Y}}$. Moreover, we define $\mathcal{G}_Z(\bar{x}, \bar{y}) \triangleq \mathcal{G}_X(\bar{x}, \bar{y}) + \mathcal{G}_Y(\bar{x}, \bar{y})$ where $\mathcal{G}_X$ is defined in Definition 2.1.

**Definition 2.3.** For a given gap function $\mathcal{G}_Z : Z \to \mathbb{R}$, a point $(\bar{x}, \bar{y}) \in Z$ is an $\epsilon$-stationary of problem (1) if $\mathcal{G}_Z(\bar{x}, \bar{y}) \leq \epsilon$.

*Remark* 2.4. As indicated in the above definitions, since we use different oracles for the maximization component of the objective function we need to employ different gap functions for the dual iterates. With a slight abuse of notation, we used $\mathcal{G}_Y$ when using both LMO and PO for the constraint set $Y$. The reason is that an $\epsilon$-solution in terms of both definitions implies an $\epsilon$-game stationary solution. More specifically, if $\mathcal{G}_Z(\bar{x}, \bar{y}) \leq \epsilon$ for some $(\bar{x}, \bar{y}) \in Z$, then

$$-\nabla_x \mathcal{L}(\bar{x}, \bar{y}) \in \partial \mathbb{1}_X(\bar{x}) + u, \quad \text{s.t. } \|u\|_{\mathcal{X}} \leq \mathcal{O}(\epsilon),$$
$$\nabla_y \mathcal{L}(\bar{x}, \bar{y}) \in \partial \mathbb{1}_Y(\bar{y}) + v, \quad \text{s.t. } \|v\|_{\mathcal{Y}} \leq \mathcal{O}(\epsilon).$$

Therefore, we consider the above definition as a unified notion of $\epsilon$-stationary solution similar to [49]. We refer the reader to [21, 28] for the details of the relation between a game stationary solution and other notations of stationary solution.

**Definition 2.5.** Let $(\mathcal{U}, \|\cdot\|_{\mathcal{U}})$ be a finite-dimensional normed vector space. A convex set $\mathcal{K} \subset \mathcal{U}$ is $\alpha$-strongly convex with respect to $\|\cdot\|_{\mathcal{U}}$ if for any $u, v \in \mathcal{K}$, $w \in \mathcal{U}$, and $\gamma \in [0, 1]$,

$$\gamma u + (1 - \gamma)v + \gamma(1 - \gamma)\frac{\alpha}{2} \|u - v\|_{\mathcal{U}}^2 \, w \in \mathcal{K}, \quad \text{s.t.} \quad \|w\|_{\mathcal{U}} = 1.$$

We next state our main assumptions considered in the paper.

## 2.2 Assumptions

**Assumption 2.6.** (I) For any $y \in Y$, $\mathcal{L}(\cdot, y)$ is continuously differentiable with a Lipschitz continuous gradient, i.e, there exists $L_{xx} \geq 0$ and $L_{yx} > 0$ such that for any $x, \bar{x} \in X$ and $y, \bar{y} \in Y$ the followings hold

$$\|\nabla_x \mathcal{L}(x, y) - \nabla_x \mathcal{L}(\bar{x}, \bar{y})\|_{\mathcal{X}^*} \leq L_{xx}\|x - \bar{x}\|_{\mathcal{X}} + L_{yx}\|y - \bar{y}\|_{\mathcal{Y}}.$$

(II) For any $x \in X$, $\mathcal{L}(x, \cdot)$ is concave and continuously differentiable with a Lipschitz continuous gradient, i.e, there exists $L_{yy} \geq 0$ and $L_{yx} > 0$ such that for any $x, \bar{x} \in X$ and $y, \bar{y} \in Y$ the followings hold

$$\|\nabla_y \mathcal{L}(x, y) - \nabla_y \mathcal{L}(\bar{x}, \bar{y})\|_{\mathcal{Y}^*} \leq L_{yx}\|x - \bar{x}\|_{\mathcal{X}} + L_{yy}\|y - \bar{y}\|_{\mathcal{Y}}.$$

(III) $X \subseteq \mathcal{X}$ and $Y \subseteq \mathcal{Y}$ are convex and compact sets with diameters $D_X$ and $D_Y$, respectively, i.e., $D_X \triangleq \sup_{x, \bar{x} \in X} \|x - \bar{x}\|_{\mathcal{X}}$ and $D_Y \triangleq \sup_{y, \bar{y} \in Y} \|y - \bar{y}\|_{\mathcal{Y}}$.

In the case where LMO is available for both minimization and maximization components of (1), we will consider the following additional assumption.

**Assumption 2.7.** $Y \subseteq \mathcal{Y}$ is $\alpha$-strongly convex for some $\alpha > 0$.

*Remark* 2.8. Strongly convex sets arise in many applications and there have been several studies characterizing various instances [15]. In particular, in many of these examples, the corresponding LMO admits a closed-form solution or can be solved efficiently. Here we present two interesting examples:

(1) Let $\mathcal{B}_f(r) \triangleq \{x \in \mathcal{X} \mid f(x) \leq r\}$ where $r > 0$ and $f : \mathcal{X} \to \mathbb{R}_+$ is a $\mu$-strongly convex and $L$-smooth function. Then, the set $\mathcal{B}_f(r)$ is strongly convex with modulus $\alpha = \mu/\sqrt{2Lr}$. In particular, this example includes $\ell_p$-norm ball when $f(x) = \|x\|_p^2$ for $p \in (1, 2]$.

(2) For a given matrix $A \in \mathbb{R}^{n \times m}$, let the singular values be denoted by $\{\sigma_i(A)\}_{i=1}^q$ where $q = \min(n, m)$. Schatten $\ell_p$ ball for $p \in (1, 2]$, i.e., $\mathcal{B}_{S(p)}(r) \triangleq \{A \in \mathbb{R}^{n \times m} \mid (\sum_{i=1}^q \sigma_i(A)^p)^{1/p} \leq r\}$, is a strongly convex set with modulus $\alpha = (p - 1)q^{\frac{1}{2} - \frac{1}{2}}/r$ (For details and more examples see [15]).

## 3 Proposed Methods

In this section, we propose our algorithms based on a primal-dual conditional-gradient approach for addressing problem (1). As previously mentioned in section 1, we assume that the minimization component of problem (1) includes a constraint set $X$, which allows for an efficient LMO, whereas the associated PO may involve computationally expensive procedures. However, with regard to the maximization component of the objective function, we consider two main scenarios based on the Oracle assumption: $(i)$ when an LMO is available; and $(ii)$ when a PO is available.

Note that problem (1) can be viewed as a minimization problem $\min_{x \in X} f(x)$ where $f(x) \triangleq \max_{y \in Y} \mathcal{L}(x, y)$. A naive implementation of a conditional gradient method, such as the Frank Wolfe method, has two main challenges: Firstly, evaluation of the objective function and/or its first-order information requires exact evaluation of $y^*(x) \in \text{argmax}_{y \in Y} \mathcal{L}(x, y)$ at each iteration given $x \in X$ which may not be possible. Secondly, since the objective function $\mathcal{L}(x, \cdot)$ is concave for any $x \in X$, it implies that $f(x)$ is a nonsmooth and nonconvex function. In fact, it can be shown that $\nabla_x \mathcal{L}(x, y^*(x)) \in \partial f(x)$, for any $y^*(x) \in \text{argmax}_{y \in Y} \mathcal{L}(x, y)$. To overcome the later challenge, a typical approach is to add a regularization term to the objective function to provide a smooth approximation for the function $f$. More specifically, one can add a regularization term $-\frac{\mu}{2} \|y - y_0\|_{\mathcal{Y}}^2$ for some $y_0 \in Y$ to the objective function leading to the following regularized SP problem

$$\min_{x \in X} \max_{y \in Y} \mathcal{L}_\mu(x, y) \triangleq \mathcal{L}(x, y) - \frac{\mu}{2} \|y - y_0\|_{\mathcal{Y}}^2.$$

Note the objective function in the above problem is strongly concave and smooth in $y$, hence, for any given $x \in X$ the corresponding maximizer $y_\mu^*(x) = \operatorname{argmax}_{y \in Y} \mathcal{L}_\mu(x, y)$ is uniquely defined. Moreover, to bypass the requirement for evaluating an exact solution at each iteration, one can provide an increasingly accurate approximated solution for $y_\mu^*(x)$, however, this leads to a two-loop method that still requires an excessive computational cost at each iteration to solve a subproblem inexactly [53]. Moreover, the performance of such inexact methods is generally sensitive to the choice of the subproblem's parameters. Therefore, to resolve these issues, we propose *single-loop* and *easy-to-implement* inexact projection-free primal-dual algorithms.

---
**Algorithm 1** Regularized Primal-dual Conditional Gradient (R-PDCG) method
---

    **Input:** $x_0 \in X$, $y_0 \in Y$, $\mu > 0$, $\{\tau_k\}_k \subseteq \mathbb{R}_+$
    **for** $k = 0, \ldots, K - 1$ **do**
        $s_k \leftarrow \operatorname{argmin}_{x \in X} \langle \nabla_x \mathcal{L}(x_k, y_k), x \rangle$
        $x_{k+1} \leftarrow \tau_k s_k + (1 - \tau_k) x_k$
        $p_k \leftarrow \operatorname{argmax}_{y \in Y} \langle \nabla_y \mathcal{L}(x_k, y_k) - \mu(y_k - y_0), y \rangle$
        $y_{k+1} \leftarrow \sigma_k p_k + (1 - \sigma_k) y_k$
    **end for**
---

In particular, when an LMO is available for both primal and dual steps, we develop an alternating conditional gradient method where at each iteration a Frank Wolfe step is taken with respect to the primal variable via the direction $\nabla_x \mathcal{L}(x_k, y_k)$ followed by a Frank Wolfe step for the regularized maximization problem with respect to the dual variable. The outline of our proposed method is presented in Algorithm 1. Moreover, considering the scenario where the projection onto set $Y$ is efficiently computable, we propose a one-sided projection-free primal-dual method. In particular, at each iteration, similar to the previous algorithm we perform a Frank Wolfe step with respect to the primal variable. Then to update the dual variable, instead of taking an FW-type update, we take a step of projected gradient ascent with respect to the regularized objective function as follows

$$y_{k+1} \leftarrow \mathcal{P}_Y \Big( y_k + \sigma_k \big( \nabla_y \mathcal{L}_\mu(x_k, y_k) \big) \Big).$$

The outline of these steps is presented in Algorithm 2.

---
**Algorithm 2** Conditional Gradient with Regularized Projected Gradient Ascent (CG-RPGA)
---

    **Input:** $x_0 \in X$, $y_0 \in Y$, $\mu > 0$, $\{\tau_k, \sigma_k\}_k \subseteq \mathbb{R}_+$
    **for** $k = 0, \ldots, K - 1$ **do**
        $s_k \leftarrow \operatorname{argmin}_{x \in X} \langle \nabla_x \mathcal{L}(x_k, y_k), x \rangle$
        $x_{k+1} \leftarrow \tau_k s_k + (1 - \tau_k) x_k$
        $y_{k+1} \leftarrow \mathcal{P}_Y \Big( y_k + \sigma_k \big( \nabla_y \mathcal{L}(x_k, y_k) - \mu(y_k - y_0) \big) \Big)$
    **end for**
---

# 4 Convergence Analysis of R-PDCG

In this section, we study the convergence properties of Algorithms 1 and 2. First, in the next lemma, we provide a one-step analysis of Algorithm 1 by providing an upper bound on the reduction of the objective function in terms of the consecutive iterates.

**Lemma 4.1.** *Suppose Assumptions 2.6 and 2.7 hold and let $\{(x_k, y_k)\}_{k \geq 0}$ be the sequence generated by Algorithm 1 with step-sizes $\tau_k = \tau > 0$ and $\{\sigma_k\}_{k \geq 0}$, and parameter $\mu > 0$. Define $\sigma_k \triangleq \min\{1, \frac{\alpha}{4(L_{yy} + \mu)} \|\nabla_y \mathcal{L}_\mu(x_k, y_k)\|_{\mathcal{Y}^*}\}$ and $H_k \triangleq \mathcal{L}_\mu(x_k, y_\mu^*(x_k)) - \mathcal{L}_\mu(x_k, y_k)$ for any $k \geq 0$. Then for any $k \geq 0$*

$$H_{k+1} \leq \max \left\{ \frac{1}{2}, 1 - \frac{\alpha \sqrt{\mu}}{8\sqrt{2}(L_{yy} + \mu)} \sqrt{H_k} \right\} H_k + \mathcal{E}(\tau), \tag{5}$$

*where $\mathcal{E}(\tau) \triangleq L_{yx} \tau D_Y D_X + 2L_{xx} \tau^2 D_X^2$.*

Note that the above one-step inequality demonstrates that given the iterate sequence $\{x_k\}_{k\geq 0}$, the generated iterates $\{y_k\}_{k\geq 0}$ reduce the suboptimality of regularized objective function $\mathcal{L}_\mu$ within an error bound $\mathcal{E}(\tau)$. In other words, the generated iterates $\{y_k\}$ provides a progressively accurate estimation of the optimal trajectory $y_\mu^*(x_k)$ with an error depending on the primal step-size $\tau$. Therefore, the key idea lies in the meticulous selection of $\tau$ to prevent error growth while ensuring sufficient progress in the primal variable. To this end, in the following theorem we establish bounds on the primal and dual gaps based on the previous lemma that are connected via parameters $\tau$ and $\mu$. Subsequently, in the next corollary by carefully selecting those parameters we demonstrate a convergence rate guarantee for Algorithm 1.

**Theorem 4.2.** *Suppose Assumptions 2.6 and 2.7 hold and let $\{(x_k, y_k)\}_{k\geq 0}$ be the sequence generated by Algorithm 1 with step-sizes $\tau_k = \tau > 0$ and $\{\sigma_k\}_{k\geq 0}$, and parameter $\mu > 0$. Define $\sigma_k \triangleq \min\{1, \frac{\alpha}{4(L_{yy}+\mu)}\|\nabla_y\mathcal{L}_\mu(x_k, y_k)\|_{\mathcal{Y}^*}\}$, then for any $K \geq 1$, there exists $t \in \{\lceil K/2 \rceil, \ldots, K-1\}$ such that $(x_t, y_t) \in X \times Y$ satisfy the following bounds*

$$\mathcal{G}_X(x_t, y_t) \leq \frac{2(f(x_0)-f(x_K))}{\tau K} + \frac{\mu}{\tau K}D_Y^2 + \frac{(L_{xx}+L_{yx}^2/\mu)\tau}{K}D_X^2 + \frac{2\sqrt{2}L_{yx}}{\sqrt{\mu}}D_X\Bigg[$$

$$\sqrt{\mathcal{E}(\tau)} + \frac{3\log(K+1)}{K}\max\left\{\sqrt{H_0}, \frac{16(L_{yy}+\mu)}{\alpha\sqrt{\mu}}\right\} + \left(\frac{8\sqrt{2}(L_{yy}+\mu)\mathcal{E}(\tau)}{\alpha\sqrt{\mu}}\right)^{1/3}\Bigg],$$

$$\mathcal{G}_Y(x_t, y_t) \leq \frac{36c_\mu}{(K+4)^2}\max\left\{H_0, \frac{256(L_{yy}+\mu)^2}{\alpha^2\mu}\right\} + c_\mu\mathcal{E}(\tau)$$

$$+ \left(\frac{8\sqrt{2}(L_{yy}+\mu)\mathcal{E}(\tau)}{\alpha\sqrt{\mu}}\right)^{2/3}c_\mu + \mu D_Y^2.$$

*where $c_\mu \triangleq 2 + L_{yy}/\mu$ and $\mathcal{E}(\tau)$ is defined in Lemma 4.1.*

Now, we are ready to state the convergence rate and complexity results of Algorithm 1.

**Corollary 4.3.** *Under the premises of Theorem 4.2, choose $\mu = \mathcal{O}(\epsilon)$ and $\tau = \mathcal{O}(\epsilon^5)$, then for any $K \geq 1$, there exists $t \in \{\lceil K/2 \rceil, \ldots, K-1\}$ such that $(x_t, y_t) \in X \times Y$ satisfy $\mathcal{G}_Z(x_t, y_t) \leq \mathcal{O}(1/K^{1/6})$. Moreover, $(x_t, y_t)$ satisfy $\mathcal{G}_Z(x_t, y_t) \leq \epsilon$ and consequently is an $\epsilon$-game stationary within $\mathcal{O}(\epsilon^{-6})$ iterations.*

Considering that the proposed method achieves convergence through dual regularization, it becomes essential to address the question of convergence guarantee for Algorithm (1) when the objective function $\mathcal{L}$ is nonconvex in $x$ and strongly concave in $y$. In such a scenario, where the objective function exhibits strong concavity, regularization can be circumvented by setting $\mu = 0$. This adjustment aligns Algorithm 1 with the approach presented in [16], albeit it should be noted that the study in [16] solely focused on convex-concave settings and their analysis does not extend to the nonconvex scenario.

**Theorem 4.4.** *Suppose Assumptions 2.6 and 2.7 hold and function $\mathcal{L}(x, \cdot)$ is $\tilde{\mu}$-strongly concave for any $x \in X$. Let $\{(x_k, y_k)\}_{k\geq 0}$ be the sequence generated by Algorithm 1 with parameter $\mu = 0$ and step-sizes $\tau_k = \mathcal{O}(1/K^{3/4})$ and $\{\sigma_k\}_{k\geq 0}$. Define $\sigma_k \triangleq \min\{1, \frac{\alpha}{4L_{yy}}\|\nabla_y\mathcal{L}(x_k, y_k)\|_{\mathcal{Y}^*}\}$, then for any $K \geq 1$, there exists $t \in \{\lceil K/2 \rceil, \ldots, K-1\}$ such that $(x_t, y_t) \in X \times Y$ satisfy:*
*(i) $\mathcal{G}_X(x_t, y_t) \leq \mathcal{O}(1/K^{1/4})$ and $\mathcal{G}_Y(x_t, y_t) \leq \mathcal{O}(1/K^{1/2})$.*
*(ii) $(x_t, y_t)$ satisfy $\epsilon$-primal gap $\mathcal{G}_X(x_t, y_t) \leq \epsilon$ within $\mathcal{O}(\epsilon^{-4})$ iterations and satisfy $\epsilon$-dual gap $\mathcal{G}_Y(x_t, y_t) \leq \epsilon$ within $\mathcal{O}(\epsilon^{-2})$ iterations.*

*Remark* 4.5. It is important to emphasize that Algorithm 1 is a single-loop method that leverages the gradient of the objective function and relies solely on the use of the LMO for handling constraints. To the best of our knowledge, our results in Corollary 4.3 and Theorem 4.4 represent the first complexity results for finding an $\epsilon$-stationary solution in nonconvex-concave and nonconvex-strongly concave SP problems, respectively, without the need for projection onto the constraint sets.

## 5 Convergence Analysis of CG-RPGA

In this section, we assume that the projection onto set $Y$ can be computed efficiently and the vector space $\mathcal{Y}$ is equipped with the Euclidean norm, i.e, $\|\cdot\|_{\mathcal{Y}} = \|\cdot\|_2$. We present the analysis for

convergence of CG-RPGA (Algorithm 2) for solving problem 1. In particular, we first consider the nonconvex-concave setting and in the following theorem and corollary, we demonstrate that the convergence rate can be improved to $\mathcal{O}(1/K^{1/4})$. Afterward, we consider a nonconvex-strongly concave setting and we establish the convergence rate results for CG-RPGA in Theorem 5.3.

**Theorem 5.1.** *Suppose Assumptions 2.6 holds and let $\{(x_k, y_k)\}_{k \geq 0}$ be the sequence generated by Algorithm 2 with step-sizes $\tau_k = \tau > 0$ and $\sigma_k = \sigma \leq \frac{2}{L_{yy}+2\mu}$, and parameter $\mu > 0$. Then for any $K \geq 1$, there exists $t \in \{\lceil K/2 \rceil, \ldots, K-1\}$ such that $(x_t, y_t) \in X \times Y$ satisfy the following bounds*

$$\mathcal{G}_X(x_t, y_t) \leq \frac{2(f(x_0) - f(x_K))}{\tau K} + \frac{\mu}{\tau K}D_Y^2 + \frac{2L_{yx}}{(1-\rho)K}D_X \left\| y_0 - y_\mu^*(x_0) \right\|_2$$
$$+ \left( \frac{2L_{yx}^2 \rho}{\mu(1-\rho)} + L_{xx} + \frac{L_{yx}^2}{\mu} \right) \tau D_X^2,$$

$$\mathcal{G}_Y(x_t, y_t) \leq \frac{2\rho^{K/2}}{\sigma} \left\| y_0 - y_\mu^*(x_0) \right\|_2 + \frac{2L_{yx}\rho}{\sigma\mu(1-\rho)}\tau D_X + \frac{L_{yx}}{\sigma\mu}\tau D_X + \mu D_Y.$$

**Corollary 5.2.** *Under the premises of Theorem 5.1, choose $\mu = \mathcal{O}(\epsilon)$ and $\tau = \mathcal{O}(\epsilon^3)$, then for any $K \geq 1$, there exists $t \in \{\lceil K/2 \rceil, \ldots, K-1\}$ such that $(x_t, y_t) \in X \times Y$ satisfy $\mathcal{G}_Z(x_t, y_t) \leq \mathcal{O}(1/K^{1/4})$. Moreover, $(x_t, y_t)$ satisfy $\mathcal{G}_Z(x_t, y_t) \leq \epsilon$ and consequently is an $\epsilon$-game stationary within $\mathcal{O}(\epsilon^{-4})$ iterations.*

Similar to section 4, it is important to address the question of convergence guarantee for Algorithm (2) when the objective function is strongly concave in $y$. In this scenario, regularization can be avoided by setting $\mu = 0$. In the following theorem, we establish the convergence rate of Algorithm 2 for SP problem (1) in this setting.

**Theorem 5.3.** *Suppose Assumptions 2.6 holds and function $\mathcal{L}(x, \cdot)$ is $\tilde{\mu}$-strongly concave for any $x \in X$. Let $\{(x_k, y_k)\}_{k \geq 0}$ be the sequence generated by Algorithm 2 with parameter $\mu = 0$ and step-sizes $\tau_k = \mathcal{O}(1/K^{1/2})$ and $\sigma_k = \sigma \leq \frac{2}{L_{yy}+\tilde{\mu}}$. Then for any $K \geq 1$, there exists $t \in \{\lceil K/2 \rceil, \ldots, K-1\}$ such that $(x_t, y_t) \in X \times Y$ satisfy $\mathcal{G}_Z(x_t, y_t) \leq \mathcal{O}(1/K^{1/2})$. Moreover, $(x_t, y_t)$ satisfy $\mathcal{G}_Z(x_t, y_t) \leq \epsilon$ within $\mathcal{O}(\epsilon^{-2})$ iterations.*

# 6 Numerical Experiment

In this section, we implement our methods to solve Robust Multiclass Classification problem described in Example 1 and Dictionary Learning problem in Example 2. For all the algorithms, the step-sizes are selected as suggested by their theoretical result and scaled to have the best performance. In particular, for R-PDCG we let $\tau = \frac{10}{K^{5/6}}$ and $\mu = \frac{10^{-3}}{K^{1/6}}$; for CG-RPGA we let $\tau = \frac{10}{K^{3/4}}$ and $\mu = \frac{10^{-3}}{K^{1/4}}$; for AGP we let the primal step-size $\frac{1}{\sqrt{k}}$, dual step-size as 0.2, and the dual regularization parameter as $\frac{10^{-1}}{k^{1/4}}$; for SPFW both primal and dual step-sizes are selected to be diminishing as $\frac{2}{k+2}$. Supplementary plots are provided in the appendix.

**Robust Multiclass Classification:** To assess the performance of our proposed algorithms (R-PDCG and CG-RPGA), we tested them against the Alternating Gradient Projection (AGP) algorithm introduced by [49] and the Saddle Point Frank Wolfe (SPFW) algorithm introduced by [16]. We conduct experiments on `rcv1` dataset ($n = 15564$, $d = 47236$, $k = 53$) and `news20` dataset ($n = 15935$, $d = 62061$, $k = 20$) from LIBSVM repository[1]. As shown in Figure 1, our algorithms outperform the competing approaches, highlighting the advantage of utilizing a projection-free approach. In this example, the high per-iteration computational cost of the projection operator significantly impacts AGP, with more than three iterations taking over 300 seconds reflecting the benefit of projection-free algorithms for a certain class of problems. For problems with easy-to-project constraints, projection-based algorithms such as AGP may have a better performance, however, this example supports the motivation behind the development of projection-free methods for saddle point problems with hard-to-project constraints, particularly when an LMO is available.

---
[1] https://www.csie.ntu.edu.tw/~cjlin/libsvmtools/datasets

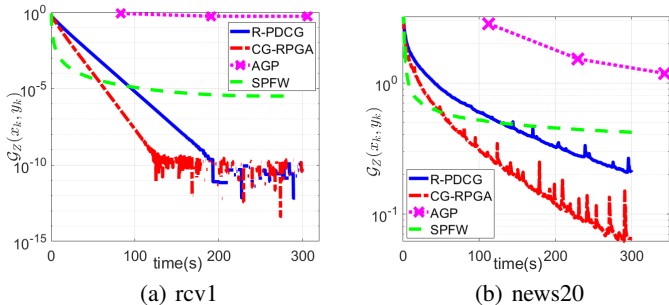

(a) rcv1                              (b) news20

Figure 1: Comparing the performance of our proposed methods R-PDCG (blue) and CG-RPGA (red) with AGP (magenta) and SPFW (green) in the Robust Multiclass Classification problem.

**Dictionary Learning:** Considering the dictionary learning problem in (4), we compared the performance of our proposed methods, R-PDCG (Algorithm 1) and CG-RPGA (Algorithm 2) with AGP [49] and SPFW [16] although SPFW does not have a theoretical guarantee for nonconvex-concave SP. The datasets are generated randomly from a standard Gaussian distribution with details described in Section F of the Appendix. Notably, CG-RPGA has a faster convergence rate compared to R-PDCG matching our theoretical results (see Table 1). Moreover, AGP which is a fully projection-based algorithm has the slowest convergence behavior in terms of time compared to other methods. Solving a linear optimization problem over the nuclear norm ball requires computing only a single pair of singular vectors corresponding to the largest singular value, whereas computing a projection onto the nuclear norm ball demands a full SVD. The computational cost of latter operation is $\mathcal{O}(kd\min(k,d))$, while the computational cost of the former one is $\mathcal{O}(\nu\ln(k+d)\sqrt{\sigma_1}/\sqrt{\epsilon})$, where $\nu \leq kd$ and $\sigma_1$ are the number of nonzero entries and the top singular value of $-\nabla_x\mathcal{L}(x,y)$, respectively, and $\epsilon$ is the accuracy [10]. Therefore, in this example, LMO is considerably more cost-effective to compute than the projection method. Figure 2 depicts our methods' superior performance compared to other algorithms.

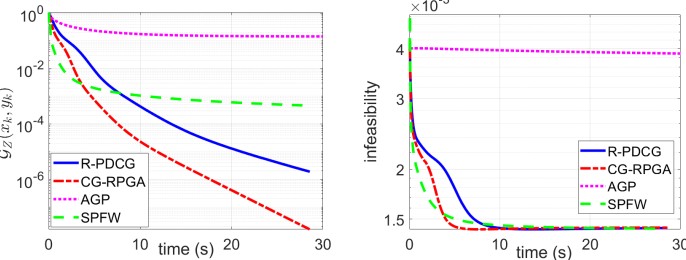

Figure 2: Comparing the performance of our proposed methods R-PDCG (blue) and CG-RPGA (red) with other methods AGP (magenta) and SPFW (green). The plots from left to right are trajectories of gap function and infeasibility for problem (4).

# 7 Conclusion

In this paper, we proposed primal-dual projection-free methods for solving a broad class of constrained nonconvex-concave problems. Using a regularization technique we devised a single-loop method relying on LMO for handling constraints. In particular, we show that R-PDCG achieves an $\epsilon$-stationary solution within $\mathcal{O}(\epsilon^{-6})$ iterations assuming that the constraint set is strongly convex. Also, our method achieves $\epsilon$-primal and $\epsilon$-dual gaps within $\mathcal{O}(\epsilon^{-4})$ and $\mathcal{O}(\epsilon^{-2})$ iterations, respectively, for nonconvex-strongly concave problems. To the best of our knowledge, this is the first fully projection-free primal-dual method with a convergence guarantee for nonconvex SP problems. Additionally, when the projection on the maximization constraint is easy to compute we propose a one-sided projection-free primal-dual method called CG-RPGA with iteration complexity of $\mathcal{O}(\epsilon^{-4})$ matching the best-known results for projection-based primal-dual methods, and improves to $\mathcal{O}(\epsilon^{-2})$ iterations for nonconvex-strongly concave setting. We acknowledge that the proposed method is currently limited to the deterministic setting and we plan to study such SP problems under uncertainty and distributed settings in future work.

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
