# OpenReview forum: "Projection-Free Methods for Solving Nonconvex-Concave Saddle Point Problems"
_NeurIPS.cc/2023/Conference — NeurIPS 2023 poster_

### Official Review · Reviewer_cP1u · 2023-06-29

**Soundness:** 3 good
**Presentation:** 2 fair
**Contribution:** 3 good
**Rating:** 5
**Confidence:** 3

**Summary:**

This paper studies the constrained nonconvex-concave minimax problem. This problem has been studied in several papers in the literature, but this paper proposes a projection free (single loop) algorithm to solve this problem.

**Strengths:**

The proposed algorithms are interesting and extend Frank-Wolf type methods to the nonconvex setting.

**Weaknesses:**

1. The authors have made some effort into explaining why the FMO oracle might be much more computationally efficient as compared to a projection onto a set. The motivating example seems to be that of the nuclear norm constraint. Can the authors describe in a little more detail as to why this is the case for this constraint. It would make the paper more complete and provide motivation to study projection free methods.

2. Once again, the motivating example of the nuclear constraint does not seem to be addressed in the assumption of $\alpha$-strongly-convex sets. Are these constraints not strongly convex? If this is the case, then there seems to be a mismatch between the motivating examples and the assumptions.

3. What is the relation between the strong convexity of the set and the strong convexity of the objective function? For example, if the inner problem is strongly concave (like in the setting of Theorem 4.4), do we still need to assume the strong convexity of the constraint set?

**Questions:**

See above

**Limitations:**

See above

---

> ### Author Rebuttal · Authors · 2023-08-10
>
> **Q1 The authors have made some effort into explaining why the FMO oracle might be much more computationally efficient as compared to a projection onto a set. The motivating example seems to be that of the nuclear norm constraint. Can the authors describe in a little more detail as to why this is the case for this constraint. It would make the paper more complete and provide motivation to study projection free methods.**
>
> **A1**
> That is an excellent question. Please note that solving a linear optimization problem over the nuclear norm ball requires computing only a single pair of singular vectors corresponding to the largest singular value, whereas computing a projection onto the nuclear norm ball demands a complete SVD decomposition. The computational cost of latter operation is $\mathcal O(kd\min(k,d))$, while the computational cost of the former one is $\mathcal O(\nu \mbox{ln}(k+d)\sqrt{\sigma_1}/\sqrt{\epsilon})$, where $\nu\leq kd$ and $\sigma_1$ are the number of nonzero entries and the top singular value of $-\nabla_x\mathcal L(x,y)$, respectively, and $\epsilon$ is the accuracy (see [R1] for more details). Therefore, in this example, LMO is considerably more cost-effective to compute than the projection method. We will add this discussion to the revised manuscript.
>
> [R1] Combettes CW, Pokutta S. Complexity of linear minimization and projection on some sets. Operations Research Letters. 2021 Jul 1;49(4):565-71.
>
> ---
>
> **Q2 Once again, the motivating example of the nuclear constraint does not seem to be addressed in the assumption of $\alpha$-strongly-convex sets. Are these constraints not strongly convex? If this is the case, then there seems to be a mismatch between the motivating examples and the assumptions.**
>
>
>
> **A2**
>  Please note that we only require the strong-convexity set assumption for the maximization problem, i.e., set $Y$. In both motivating examples, the nuclear norm constraint is used for the minimization part of the objective function, where we do not require strong convexity of the constraint set assumption. More specifically, the Continual Dictionary Learning example in our paper effectively exhibits the application of strong convexity of the set $Y$ as it includes an $\ell_2$-norm ball constraint.
>  Additionally, for the Robust Multiclass Classification example, the constraint of the maximization is the intersection of simplex set and divergence measure constraints. Indeed, one can relax the simplex constraint using the splitting technique and Fenchel duality. The resulting equivalent saddle point problem has a maximization constraint of $Y=${$y:V(y,\frac{1}{n}\mathbf 1_n)\leq \rho$}
> which is only described by the divergence measure constraint. In some popular examples such as the Pearson Chi-square divergence, i.e., $V(y,\mathbf{1}_n/n)=||ny-\mathbf{1}_n||^2$, $Y$ satisfies the assumption of strongly convex constraint set. We will add a more detailed discussion in this regard to the revised manuscript.
>
> ---
>
> **Q3 What is the relation between the strong convexity of the set and the strong convexity of the objective function? For example, if the inner problem is strongly concave (like in the setting of Theorem 4.4), do we still need to assume the strong convexity of the constraint set?**
>
>
> **A3**
> The definitions of 'strongly convex set' and 'strongly convex objective function' are distinct from each other. For Algorithm R-PDCG, even in strongly-concave setting, we still need the strong convex set for the maximization problem. This assumption plays a critical role in our convergence analysis which is closely related to the analysis of FW-type methods.
>
> To gain a deeper understanding of this assumption, it is helpful to examine the convergence results of FW-type methods for solving strongly convex minimization problems. Classical studies on FW-type methods have shown that, unlike projection-based methods, strong convexity of the objective function does not necessarily lead to an accelerated rate (faster than $\mathcal O(1/K)$) [R2]. Achieving a faster rate often requires imposing additional assumptions, such as the existence of a solution in the interior of the domain or a uniform lower bound on the norm of the gradient of the objective function at the solution.
> It is important to note that the extension of most of these assumptions to the min-max problems may not lead to a reasonable assumption since the solution set corresponding to the maximization problem $\mathcal Y^\star(x)=\hbox{argmax}_{y\in Y}\mathcal L(x,y)$ as well as the gradient at the maximizer $\nabla_y\mathcal L(x,y^\star(x))$ changes with respect to $x$. Consequently, our novel analysis using the relatively mild set of assumptions led to convergence results that appeared for the first time in the literature for the considered setting.
>
> [R2] Lan G, Zhou Y. Conditional gradient sliding for convex optimization. SIAM Journal on Optimization. 2016;26(2):1379-409.

---

> > ### Comment · Reviewer_cP1u · 2023-08-15
> > **Response to Rebuttal**
> >
> > I'd like to thank the authors for their response. I have increased my score.

---

### Official Review · Reviewer_LwGk · 2023-07-02

**Soundness:** 3 good
**Presentation:** 3 good
**Contribution:** 2 fair
**Rating:** 5
**Confidence:** 3

**Summary:**

This paper investigates algorithms for constrained saddle point (SP) problems where the objective function is nonconvex-concave and smooth. Existing methods are usually projection-based and this paper focuses on developing single-loop projection-free algorithms which only use linear minimization oracles.

In particular, this paper provides convergence guarantees for nonconvex-concave SP problems and nonconvex-strongly concave SP problems. This paper also investigates one-sided projection-free methods which can achieve an improved convergence matching the SOTA results for projection-based methods.

**Strengths:**

This paper is well-written and easy to follow. The contributions of this paper are also straightforward: developing projection-free algorithms for saddle point problems and providing convergence guarantees for the proposed methods. By considering the LMO-PO oracle, the rate obtained matches the SOTA convergence of projection-based algorithms.

**Weaknesses:**

1) When analyzing the convergence guarantees for the R-PDCG method, the author also assumes that Y is S-Convex set. Is this condition inevitable? Also, the rate for R-PDCG is slightly worse than projection-based methods, is this rate improvable, or it is already optimal for projection-free algorithms?
2) It seems a bit strange that the experiment results are shown in the Introduction part. The paper would be more convincing if the authors can present more extensive experiments and additional numerical results.
3) It seems that parameter $\mu$ and $\tau$ are related to parameter $K$; how to set $K$ in practice?

**Questions:**

See the "Weaknesses" part.

**Limitations:**

The authors addressed the limitations adequately.

---

> ### Author Rebuttal · Authors · 2023-08-10
>
> **Q1 When analyzing the convergence guarantees for the R-PDCG method, the author also assumes that Y is S-Convex set. Is this condition inevitable? Also, the rate for R-PDCG is slightly worse than projection-based methods, is this rate improvable, or it is already optimal for projection-free algorithms?**
>
>
> **A1**
> Thank you for raising this question.
> We would like to remark that the lower bound complexity for finding an $\epsilon$-stationary of the problem (1) in nonconvex-concave setting is not known yet. However, the complexity of our proposed algorithm CG-RPGA matches with the state-of-the-art single-loop algorithms (see [13]).
> On the other hand, classical studies on FW-type methods have shown that, unlike projection-based methods, strong convexity of the objective function does not necessarily lead to an accelerated rate (faster than $\mathcal O(1/K)$) [R1]. Therefore, we conjecture that our complexity result for R-PDCG is indeed optimal. Achieving a faster rate often requires imposing additional assumptions (see [31]), such as the existence of a solution in the interior of the domain or a uniform lower-bound on the norm of the gradient of the objective function at the solution.
> It is important to note that the extension of most of these assumptions to the min-max problems may not be reasonable since the maximization problem is parameterized by the minimizer variable $x$. Consequently, our novel analysis using the relatively mild set of assumptions led to convergence results that appeared for the first time in the literature for the considered setting.
>
> [R1] Lan G, Zhou Y. Conditional gradient sliding for convex optimization. SIAM Journal on Optimization. 2016;26(2):1379-409.
>
> ---
>
> **Q2 It seems a bit strange that the experiment results are shown in the Introduction part. The paper would be more convincing if the authors can present more extensive experiments and additional numerical results.**
>
>
>
> **A2**
> Thank you for your suggestions. We will add more experiments to the revised manuscript. Specifically, we have implemented our proposed algorithms to address the Robust Multiclass Classification example and have compared the results with those of competitive schemes. As evident in the attached PDF file, our methods outperform the others, highlighting the advantage of utilizing a projection-free approach. Furthermore, as suggested by the reviewer, we will incorporate a ``Numerical Experiments" section in the revised manuscript and relocate the plots to this section.
>
> ---
>
> **Q3 It seems that parameter
>  $\mu$ and $\tau $ are related to parameter $K$; how to set $K$
>  in practice?**
>
>
> **A3**
> Thank you for bringing up this point. Please note that the step-size $\tau_k$ and parameter $\mu_k$ can be equivalently selected in terms of the user-specified accuracy $\epsilon>0$. Specifically, in Corollary 4.3, we have $\mu=\mathcal O(\epsilon)$ and $\tau=\mathcal O(\epsilon^5)$. In Corollary 5.2, we find $\mu=\mathcal O(\epsilon)$ and $\tau=\mathcal O(\epsilon^3)$. Therefore, these parameters can be set according to the user-prescribed parameter $\epsilon$. We will modify these parameter selections in the Corollaries accordingly for clarification.

---

> > ### Comment · Reviewer_LwGk · 2023-08-18
> >
> > Thank you for your response! I have increased my score.

---

### Official Review · Reviewer_dDgV · 2023-07-04

**Soundness:** 4 excellent
**Presentation:** 4 excellent
**Contribution:** 2 fair
**Rating:** 6
**Confidence:** 4

**Summary:**

This paper proposes projection-free optimization algorithms for constrained nonconvex-(strongly) concave saddle point problem.

Solution concept: $\epsilon$-stationarity

To this end, they propose R-PDCG and CG-RPGA algorithms.
1. Without projection, iteration complexity for R-PDCG is $O(\epsilon^{-6})$ for nonconvex-concave and  $O(\epsilon^{-4})$ nonconvex- strongly concave objective.

2. With projection for the maximization step, iteration complexity for CG-RPGA is $O(\epsilon^{-4})$ for nonconvex-concave and  $O(\epsilon^{-2})$ nonconvex- strongly concave objective.

They illustrate the results through experiments.


**Strengths:**

1. The problem is well-motivated and the applications are clear.
2. This is the first projection-free method for constrained nonconvex-concave saddle point problem.

**Weaknesses:**

I am judging the paper by it's theoretical contribution because the applications are supportive of the theory but that's not the main message.


The main weakness of the work is the novelty of the theoretical tools used in the proofs. **Lemma 4.1 + Lemma B.1 are the keys to the proofs.
The proof techniques, specifically (12)-(14) (rest of the proof follows by assumptions) are standard steps in any FW-based method for smooth functions (see [14] for example).**


**Questions:**


1. Could you comment on the optimality of the bounds, i.e., how tight are the bounds?

2. Could you highlight the novelties required for the proof beyond Lemma 4.1, and Lemma B.1?

3. In the experiments, the algorithm does converge to a stationary point, but does stationarity guarantee a good solution for these applications?

4. It seems like if the function is smooth nonconvex-nonconcave, the algorithm should guarantee stationarity. If yes, you should highlight that in the paper. If not, could you explain why? In other words, how important is the convexity?

**Limitations:**

See "Weakness" section.

---

> ### Author Rebuttal · Authors · 2023-08-10
>
> **Q1 Could you comment on the optimality of the bounds, i.e., how tight are the bounds?**
>
> **A1**
> Thank you for raising this question. We would like to remark that the lower bound complexity for finding an $\epsilon$-stationary of the problem (1) in nonconvex-concave setting is not known yet. However, we comment that the complexity of our proposed algorithm CG-RPGA matches with the state-of-the-art single-loop algorithms (see [13]). Moreover, considering the nonconvex-strongly concave setting, our proposed methods match the lower bound complexity of $\mathcal O(\epsilon^{-2})$ (see [R1]).
>
> [R1] Li H, Tian Y, Zhang J, Jadbabaie A. Complexity lower bounds for nonconvex-strongly-concave min-max optimization. Advances in Neural Information Processing Systems. 2021 Dec 6;34:1792-804.
>
> ---
>
> **Q2 Could you highlight the novelties required for the proof beyond Lemma 4.1, and Lemma B.1? The proof techniques, specifically (12)-(14) (rest of the proof follows by assumptions) are standard steps in any FW-based method for smooth functions (see [14] for example).**
>
>
>
> **A2** That's a great question.
> The novelty of our methods lies in addressing the maximization component of the problem as a parametric optimization problem: $\max_{y\in Y} \mathcal L(x,y)$. Our convergence analysis goes beyond established methods, showcasing new and valuable insights. Notably, our analysis introduces new intermediary steps, exemplified by Lemma 4.1. Within this lemma, we illustrate how the suboptimality of the regularized objective function $\mathcal L_\mu(x_k,\cdot)$ can be diminished within the bound of the error term $\mathcal E(\tau)$. These new steps lead to new results that appear for the first time in the literature (see Theorem 4.2 and 5.1). Although some of the analysis steps resemble the standard analysis of FW-type methods, we present unique inequalities. An illustration of this is equation (17), which emerges as a consequence of proving the technical Lemma B.1. This enriches the scope of our findings and distinguishes our work in this domain.
>
> ---
>
> **Q3 In the experiments, the algorithm does converge to a stationary point, but does stationarity guarantee a good solution for these applications?**
>
>
> **A3**
> Thank you for sharing your feedback. In our paper, all of our proposed methods achieve an $\epsilon$-game stationary gap. The relation between $\epsilon$-game stationary and other $\epsilon$-stationary points of saddle point problems have been studied extensively in [R2]. For instance, in the Dictionary Learning example considered in our paper an $\epsilon$-game stationary solution leads to an $\epsilon$-infeasibility and reduction of the objective loss function. This observation aligns perfectly with our primary goal of Dictionary Learning.
>
> [R2] Li J, Zhu L, So AM. Nonsmooth Composite Nonconvex-Concave Minimax Optimization. arXiv preprint arXiv:2209.10825. 2022 Sep 22.
>
> ---
>
> **Q4 It seems like if the function is smooth nonconvex-nonconcave, the algorithm should guarantee stationarity. If yes, you should highlight that in the paper. If not, could you explain why? In other words, how important is the convexity?**
>
>
> **A4**
> We are a bit puzzled by the reviewer's question. It would be great if the reviewer can provide more details about the question. The definition of the stationary solution is presented in Definition 2.3 of the paper and we provide a gap function in Definitions 2.1 and 2.2 to measure an $\epsilon$-stationary solution of the saddle problem. Due to the lack of convexity assumption, our only hope is to guarantee a stationary solution which is stated in the results of the paper (see Corollary 4.3 and 5.2).

---

> ### Comment · Reviewer_dDgV · 2023-08-19
> **Thanks for the response**
>
> I am happy with the response and keep my score.

---

### Official Review · Reviewer_WRhg · 2023-07-06

**Soundness:** 3 good
**Presentation:** 3 good
**Contribution:** 2 fair
**Rating:** 6
**Confidence:** 3

**Summary:**

This paper proposed two projection-free algorithms for solving smooth nonconvex- (strongly) concave saddle point problems. The authors showed that the convergence rates of the proposed algorithms matches the state-of-the-art convergence rate of projection-based methods. Experimental results on dictionary learning verify that the proposed algorithms show great advantage in terms of training time compared to existing projection-based methods.

**Strengths:**

The motivation and contribution of this paper are clear. Projections in algorithms could be problematic in practice and potentially slow down the training. This work fills this gap in nonconvex-(strongly) concave saddle point problems.

**Weaknesses:**

I do not see any major weakness, but I do have some questions. Please see the Questions section.

**Questions:**

1. In the related work section, I would recommend the authors to discuss the works on projection-free methods for solving bilevel optimization. Since saddle point problem can be viewed as a special case of bilevel problem, methods proposed for solving bilevel problems should be applicable for SP problems.

2. The result in line 247 needs a reference. I believe it is related to Danskin's Lemma.

3. In the experiment part, I noticed that the one-sided projection-free method CG-RPGA has better performance in terms of training time compared with the fully projection-based method R-PDCG. The authors argue that this matches the convergence rates. I'm not sure about this argument, because the convergence rates are in terms of number of iterations and time per-iteration is obviously different for each methods. I think the training time plots are trying to verify that even AGP has the same convergence rate as CG-RPGA, AGP is slower mainly due to the projection oracle. If the authors would like to verify the different convergence rates of CG-RPGA and R-PDCG, a plot of gap function and iteration number is more appropriate.

**Limitations:**

Limitations are well discussed in the last section of the paper.

---

> ### Author Rebuttal · Authors · 2023-08-10
>
> **Q1 It is recommended that the authors discuss the works on projection-free methods for solving bilevel optimization.**
>
>
> **A1** Thanks for the great suggestion. There is indeed a connection between bilevel optimization and saddle point (SP) problems and we will add the related work on bilevel optimization in the revised manuscript. However, it is important to note that most of the existing methods for solving bilevel optimization problems consider an additional assumption on the lower-level objective function satisfying strong convexity or Polyak {\L}ojasiewicz (PL) condition. These assumptions in the context of SP problems translate into strong concavity or PL condition for $\mathcal L(x,\cdot)$ which cannot handle merely concave setting considered in this paper to the best of our knowledge.
>
> ---
>
> **Q2 The result in line 247 needs a reference. I believe it is related to Danskin's Lemma.**
>
> **A2** Thanks for pointing this out. We will add the reference in the revised manuscript.
>
> ---
>
> **Q3 In the experiment part, I noticed that the one-sided projection-free method CG-RPGA has better performance in terms of training time compared with the fully projection-based method R-PDCG. The authors argue that this matches the convergence rates. I'm not sure about this argument, because the convergence rates are in terms of number of iterations and time per-iteration is obviously different for each methods. I think the training time plots are trying to verify that even AGP has the same convergence rate as CG-RPGA, AGP is slower mainly due to the projection oracle. If the authors would like to verify the different convergence rates of CG-RPGA and R-PDCG, a plot of gap function and iteration number is more appropriate.**
>
> **A3**
> We believe that there is confusion regarding the algorithms plots and their convergence rate which we would like to clarify. Note that R-PDCG has a complexity of $\mathcal O(1/\epsilon^6)$ while CG-RPGA has a complexity of $\mathcal O(1/\epsilon^4)$. From Figure 1 in the paper, it can be observed that CG-RPGA has a faster convergence compared to R-PDCG which matches the complexity results obtained in the paper. Moreover, we believe that the plots in terms of time demonstrate a better picture of comparing these methods. Note that one of the main goals of our paper is to show the advantage of using LMO for certain classes of problems. This is indeed the case when the computational cost of LMO is cheaper than PO which can be observed when comparing the computational cost of these algorithms. Per the reviewer's suggestion, the plots of the algorithms in terms of iteration counters will be added to the paper (see the attached pdf file).

---

> > ### Comment · Reviewer_RF27 · 2023-08-15
> >
> > In the plots of the algorithms in terms of iteration counters in Example 2 (Dictionary Learning),  I wonder why the fully projection-based algorithm AGP has a slower convergence performance than FW-based algorithms. It seems that AGP has a comparable convergence result to the result of CG-RPGA and is even faster than R-PDCG in theory.

---

> > > ### Author Response · Authors · 2023-08-15
> > >
> > > We appreciate the reviewer for the follow-up question. We would like to highlight that in the examples we considered in the paper, the projection onto the constraint set $X$ requires full SVD decomposition, therefore, it leads to a higher computational cost for AGP algorithm. As we fixed the running time of algorithms in all the experiments, AGP will take fewer iterations compared to other methods. Therefore, it is very important to acknowledge the distinct oracles these algorithms employ when comparing their complexity results as stated in Table 1 of our paper.  Moreover, it should be noted that the complexity results available for these methods are only upper bounds for the gap functions and the algorithms may have better performances on specific examples. The examples provided in the numerical experiments support the motivation behind the development of projection-free methods for saddle point problems, particularly when an LMO is available.

---

> > ### Comment · Reviewer_WRhg · 2023-08-15
> >
> > Thanks the authors for the response and additional plots.
> >
> > For the Robust Multiclass Classification experiment, due to the few iterations of AGP shown in the iteration plots in Figure 1 and Figure 2, it is still not clear how AGP performs compared with the proposed methods in terms of iterations. Moreover, as Reviewer RF27 mentioned in the comment, in the Dictionary Learning experiment, AGP converges much slower in terms of iterations. This is does not match the theory. I wonder if the authors have any insights on this observation.
> >
> > For now, I will keep my score unchanged.

---

> > > ### Author Response · Authors · 2023-08-15
> > >
> > > This aligns precisely with the core motivation of our paper. As detailed in our response to Reviewer cP1u, projecting onto the nuclear-norm constraint incurs a higher computational cost compared to the corresponding LMO. The reason is that the projection operation requires a full SVD decomposition while LMO requires finding the left and right singular vectors corresponding to the largest singular value of $\nabla_x\mathcal L(x_k,y_k)$ (Please also see our response to reviewer RF27). In the Robust Multiclass Classification example, we observe that the high per-iteration computational cost of the projection operator significantly impacts AGP, with more than three iterations taking over 300 seconds reflecting the benefit of projection-free algorithms for a certain class of problems. Therefore, it is very important to acknowledge the distinct oracles these algorithms employ when comparing their complexity results as stated in Table 1 of our paper. For problems with easy-to-project constraints, projection-based algorithms such as AGP may have a better performance, however,
> > > the examples provided in the numerical experiments support the motivation behind the development of projection-free methods for saddle point problems with hard-to-project constraints, particularly when an LMO is available. For the final version of the paper, we will run AGP for additional iterations to enhance the clarity of its performance.
> > >
> > > We appreciate the reviewer's question and would be more than happy to address any other concerns they may have.

---

### Official Review · Reviewer_RF27 · 2023-07-26

**Soundness:** 3 good
**Presentation:** 3 good
**Contribution:** 3 good
**Rating:** 7
**Confidence:** 4

**Summary:**

This paper proposes two Frank-Wolfe (FW) based algorithms for solving a class of nonconvex-(strongly) concave saddle point problems. The proposed algorithms are among the first projection-free methods with convergence guarantees for such problems as authors claimed. The paper uses regularization and nested approximation techniques to deal with the nonsmooth component, and apply it to the primal-dual scheme.  Shortly, it approximates the nonsmooth function $f(x)$ by $\mu$ in convex-concave setting.  If the objective function is strongly concave in $y$, the regularization can be avoided by setting $\mu=0$. In terms of novelty, the techniques used in this paper are common, and the analysis seems quite classic to me. However, this simple combination still brings interesting results. I think this is a very good and well written paper, and it has made a great contribution. Therefore, I suggest accepting this paper, but I may change my perspective based on other comments.


**Strengths:**

Originality. This paper is a good combination of the regularization technique and Frank-Wolfe method. This allows to obtain a single-loop projection-free method with cheaper computational cost for the nonconvex-concave problem.

Quality. As far as I see, the proofs are correct. Experiments show the advantage of the projection-free methods. It would be better to add another simulation example for situations where projecting on constraints $X$ and $Y$ are difficult.

Clarity. The paper is easy to understand and the results are clearly stated and well-organized. I would like to suggest the authors to double check language, symbols, and definitions. For example, "problem 1" should be changed to "problem (1)", and $\mathcal{G}_X(\bar z)$ should be changed to $\mathcal{G}_X(\bar x,\bar y)$ in Definition 2.1.

Significance.  This paper considers a class of nonconvex-concave saddle-point problems, which widely exists in robust optimization, reinforcement learning and adversarial learning. Given existing results, the main contribution of this paper are about solving such problems via projection-free schemes, which reduce the computational complexity in dealing with the problem with structured complicated constraint set, such as nuclear norm ball.  The proposed methods can be useful in practice because of its cheaper computational cost and ability to solve the problem with complicated constraint sets.

**Weaknesses:**

The convergence requirement of the fully projection-free method R-PDCG  is that the set $Y$ is strongly convex, which is very limited in practical applications. If this assumption can be removed while achieving faster convergence performance (comparable to projection based methods), it would be a better result. In addition, the value of step size $\tau_k$ and the parameter $\mu_k$ is related to the total iteration $K$. If the total number of iterations is large, this will result in a small step size of the algorithms and slow convergence. It would be better to improve the step size $\tau_k$ to a constant that is independent of the total number of iterations.


**Questions:**

The four theorems proposed in this paper said that "there exists $t\in\{\cdots\}$ such that ... satisfy the following bounds". Does this mean that only a limited amount of iterations satisfies the boundary? Is this measure reasonable and what is its practical significance?

---

> ### Author Rebuttal · Authors · 2023-08-10
>
> **Q1
> Add an example where projecting on constraints are difficult**
>
> **A1**
> Thank you for your suggestions. We will add more experiments to the revised manuscript. Specifically, we have implemented our proposed algorithms to address the Robust Multiclass Classification example and have compared the results with those of competitive schemes. (see the attached PDF file)
>
> ---
>
> **Q2 The convergence of R-PDCG required the set $Y$ to be strongly convex, which is very limited in practical applications. Can this assumption be removed?**
>
> **A2**
> The strong convexity for set $Y$ plays a critical role in our convergence analysis which is closely related to the analysis of FW-type methods.
>
> To gain a deeper understanding of this assumption, it is helpful to examine the convergence results of FW-type methods for solving strongly convex minimization problems. Classical studies on FW-type methods have shown that, unlike projection-based methods, strong convexity of the objective function does not necessarily lead to an accelerated rate (faster than $\mathcal O(1/K)$) [R1]. Achieving a faster rate often requires imposing additional assumptions (see [31]), such as the existence of a solution in the interior of the domain or a uniform lower-bound on the norm of the gradient of the objective function at the solution.
> It is important to note that the extension of most of these assumptions to the min-max problems may not be a reasonable assumption since the solution set corresponding to the maximization problem $\mathcal Y^\star(x)=\hbox{argmax}_{y\in Y}\mathcal L(x,y)$ as well as the gradient at the maximizer $\nabla_y\mathcal L(x,y^\star(x))$ changes with respect to $x$. Consequently, our novel analysis using the relatively mild set of assumptions led to convergence results that appeared for the first time in the literature for the considered setting.
>
> Our motivating examples along with Remark 2.8 underscores the significant relevance of our assumption in various machine learning applications. In Section 5 of reference [31], a comprehensive explanation of diverse examples and their applications is provided which further emphasizes the practical implications of our work.  We would also like to mention that the applications of strongly convex set even goes beyond machine learning and it has also a subject of interest in optimal control theory [R2].
>
> In our paper, we show two specific applications satisfying the strongly convex set assumption. The Continual Dictionary Learning example in our paper effectively exhibits the application of strong convexity of the set $Y$ as it includes an $\ell_2$-norm ball constraint. For the Robust Multiclass Classification example, the constraint of the maximization is the intersection of simplex set and divergence measure constraints. Indeed, one can relax the simplex constraint using the splitting technique and Fenchel duality. The resulting equivalent saddle point problem has a maximization constraint of $Y=${$y:V(y,\frac{1}{n}\mathbf 1_n)\leq \rho$}.
> which is only described by the divergence measure constraint. In some popular examples such as the Pearson Chi-square divergence, i.e., $V(y,\mathbf{1}_n/n)=||ny-\mathbf{1}_n||^2$, $Y$ satisfies the assumption of strongly convex constraint set. We will add a more detailed discussion in this regard to the revised manuscript.
>
> [R1] Lan G, Zhou Y. Conditional gradient sliding for convex optimization. SIAM Journal on Optimization. 2016;26(2):1379-409.
>
> [R2] Veliov VM, Vuong PT. Gradient methods on strongly convex feasible sets and optimal control of affine systems. Applied Mathematics \& Optimization. 2020 Jun;81:1021-54.
>
> ---
>
> **Q3 The value of $\tau_k$ and $\mu_k$ is related to the total iteration $K$.**
>
> **A3**
> Please note that the step-size $\tau_k$ and parameter $\mu_k$ can be equivalently selected in terms of the user-specified accuracy $\epsilon>0$. Specifically, in Corollary 4.3, we have $\mu=\mathcal O(\epsilon)$ and $\tau=\mathcal O(\epsilon^5)$. In Corollary 5.2, we find $\mu=\mathcal O(\epsilon)$ and $\tau=\mathcal O(\epsilon^3)$. Therefore, these parameters can be set according to the user-prescribed parameter $\epsilon$. We will modify these parameter selections in the Corollaries accordingly for further clarification.
>
> ---
>
> **Q4 The theorems said that "there exists such that $t\in ...$ satisfy the following bounds". Is this measure reasonable and what is its practical significance?**
>
> **A4**
> Thank you for raising this question. In our convergence analysis, we demonstrated that after performing $K$ iterations, our proposed methods guarantee that at least one of the iterations in $\{(x_k,y_k)\}_{k=1}^{K}$, say $(x_t,y_t)$, satisfies $\epsilon$-gap criterion, i.e., $\mathcal G_Z(x_t,y_t)\leq \epsilon$.
> We highlight that this criterion is indeed easy to track during the course of the algorithm. In particular, one can track the values $\mathcal G_Z(x_k,y_k)=\langle \nabla_x \mathcal L(x_k,y_k),x_k-s_k\rangle + \langle \nabla_y \mathcal L(x_k,y_k),p_k-y_k\rangle$ without any additional cost at each iteration until it reaches or falls below the desired accuracy $\epsilon>0$.

---

> > ### Comment · Reviewer_RF27 · 2023-08-15
> >
> > It would be better to provide the step sizes of different algorithms used in Simulation. In addition, please state the reason for the choice of $\tau_k$ and $\mu_k$ , which are not discussed in the whole paper.

---

> > > ### Author Response · Authors · 2023-08-15
> > >
> > > Thank you for pointing this out.
> > >
> > > (i) Due to space limitations, we have relegated the details of our experiment to section F of the Appendix. In the final version of the paper, having one additional page will allow us to relocate this information to the main body of the paper. As it is mentioned in Appendix section F, ``For all the algorithms, the step-sizes are selected as suggested by the papers and scaled to have the best performance. For AGP we let the primal step-size $\frac{1}{\sqrt{k}}$, dual step-size as 0.2, and the dual regularization parameter as $\frac{10^{-1}}{k^{1/4}}$; for SPFW both primal and dual step-sizes are selected to be diminishing as $\frac{2}{k+2}$".
> > >
> > > (ii) The selection of the step size $\tau_k$ and parameter $\mu_k$ is discussed in the proof of Corollaries 4.3 and 5.2. These parameters are selected to minimize the upper bound derived in Theorems 4.2 and 5.1, respectively. For instance, in Theorem 4.2, we have provided an explicit upper bound on the primal and dual gap functions. Considering the dominant terms in the aggregation of these two bounds in terms of $\tau,\mu$, and $K$ we observe that $\mathcal{G}_Z(x_t,y_t)\leq \mathcal O(\frac{1}{\tau K}+\frac{\tau^{1/3}}{\mu^{2/3}}+\mu)$. Therefore, selecting $K=\mathcal O(\epsilon^{-6})$, $\tau=\mathcal O(\epsilon^5)$ and $\mu=\mathcal O(\epsilon)$ implies that $\mathcal{G}_Z(x_t,y_t)\leq \epsilon$ after $K=\mathcal O(\epsilon^{-6})$ iterations.

---

> > > > ### Comment · Reviewer_RF27 · 2023-08-16
> > > >
> > > > Many thanks for the authors' response.
> > > >
> > > > I strongly agree with the author's statement that projection-free methods may exhibit better convergence performance than projection-based methods in terms of the running time under some special set constraints. But I think  the projected method should exhibit comparable or better performance than the projection-free method in terms of the iteration in theory even in hard-to-project cases, which is inconsistent with the first plot (left figure) of the second example. Could the author explain the reason why the FW-based algorithms  exhibit  faster convergence performance than AGP in terms of iteration in the second example?
> > > >
> > > > Also, I have one more question. In the simulation,  what method was used to solve the projection operation of the projected-based algorithm AGP, especially in the case of the constraint set being a nuclear norm ball, whether there are differences in the algorithm performance caused by different solution methods.

---

> > > > > ### Author Response · Authors · 2023-08-16
> > > > >
> > > > > (i) Thank you for your follow-up questions.  One important property of FW-type methods is that they usually add just one point (often a vertex) in each iteration. This is an important property in many applications For example, when the optimization has a nuclear norm ball constraint each step adds only one rank-1 matrix and it leads to a sparse iterates. After $t$ iterations, the resulting solution has a rank of at most $t+1$. A reason that our methods outperform AGP in these examples can stem from this important property of FW-type methods as the optimal (stationary) solutions are potentially sparse. See section 1.1. of [R1] for more details.
> > > > >
> > > > > To have a better understanding of AGP performance, we re-executed the method with tuned stepsizes, resulting in a comparable outcome for AGP when compared to R-PDCG. In the final version of the paper, we will incorporate new plots showcasing the performance of AGP. Additionally, for the Robust Multiclass Classification problem, we intend to conduct further AGP iterations to provide enhanced clarity regarding its performance.
> > > > >
> > > > > (ii) The primal projection step in AGP involves projecting onto nuclear-ball constraint for variable $C'$ and projecting onto column-wise unit $l2$-norm ball for variable $D'$. For projection onto nuclear-ball constraint, we invoked svd function in Matlab to decompose the gradient at each iteration, and subsequently the vector of singular values is projected onto the simplex set.
> > > > >
> > > > > [R1] Braun, G., Carderera, A., Combettes, C. W., Hassani, H., Karbasi, A., Mokhtari, A., \& Pokutta, S. (2022). Conditional gradient methods. arXiv preprint arXiv:2211.14103.

---

> > > > > > ### Comment · Reviewer_RF27 · 2023-08-18
> > > > > >
> > > > > > Thanks for the authors' response. I think it is a solid paper. I will keep my score unchanged.

---

### Author Rebuttal · Authors · 2023-08-10

In response to the questions from reviewers, we have implemented our proposed algorithms to address the Robust Multiclass Classification example and have compared the results with those of competitive schemes. Moreover, in response to the reviewer WRhg, for the Dictionary Learning problem, the plot of the algorithms in terms of iteration counters is added.

---

### Decision · Program_Chairs · 2023-09-21

**Decision:**

Accept (poster)

**Comment:**

The paper investigates projection-free methods for a class of nonconvex-concave saddle point problems. Two methods are proposed: The first one (R-PDCG) employs FW-type steps (LMO-based) for both minimization and maximization constraints. This method requires the constraint set for the maximization problem to be strongly convex and finds an eps-stationary solution to the saddle point problem in O(eps^{-6}) iterations. The second method (CG-RPGA) employs FW-steps for minimization but projection-based steps for the maximization problem and achieves an eps-stationary solution in O(eps^{-4}) iterations. The rates improve for nonconvex-stronglyconcave problems.

The reviewing team has collectively voted that the paper surpasses the acceptance threshold. The problem at hand is well-motivated, and the results presented are considered novel. The NeurIPS community has exhibited interest in both FW-type methods and saddle point problems in recent years. Therefore, the exploration of projection-free methods for nonconvex-concave saddle point problems aligns naturally. On the weaker side, the primary limitation lies in the requirement of a strongly convex set for the fully projection-free variant (referred to as R-PDCG). Several reviewers have highlighted this as a restrictive assumption in practical scenarios. Although the authors reference (Garber & Hazan, 2015) for a list of examples where this assumption holds, I agree with the reviewers that the number of applications meeting the criteria of a strongly convex constraint set, easy-to-solve linear minimization, and challenging projection is somewhat limited. Whether similar guarantees can be achieved without a strong convexity assumption on the constraints remains an interesting open problem.